# Exercise Strengthens Central Nervous System Modulation of Pain in Fibromyalgia

**DOI:** 10.3390/brainsci6010008

**Published:** 2016-02-26

**Authors:** Laura D. Ellingson, Aaron J. Stegner, Isaac J. Schwabacher, Kelli F. Koltyn, Dane B. Cook

**Affiliations:** 1Department of Kinesiology, Iowa State University, Ames, IA 50011, USA; 2William S. Middleton Memorial Veterans Hospital, Madison, WI 53706, USA; astegner@wisc.edu (A.J.S.); ischwabacher@wisc.edu (I.J.S.); dane.cook@wisc.edu (D.B.C.); 3Department of Kinesiology, University of Wisconsin-Madison, Madison, WI 53706, USA; kelli.koltyn@wisc.edu

**Keywords:** modulation, exercise, chronic pain, imaging, fibromyalgia

## Abstract

To begin to elucidate the mechanisms underlying the benefits of exercise for chronic pain, we assessed the influence of exercise on brain responses to pain in fibromyalgia (FM). Complete data were collected for nine female FM patients and nine pain-free controls (CO) who underwent two functional neuroimaging scans, following exercise (EX) and following quiet rest (QR). Brain responses and pain ratings to noxious heat stimuli were compared within and between groups. For pain ratings, there was a significant (*p* < 0.05) Condition by Run interaction characterized by moderately lower pain ratings post EX compared to QR (*d* = 0.39–0.41) for FM but similar to ratings in CO (*d* = 0.10–0.26), thereby demonstrating that exercise decreased pain sensitivity in FM patients to a level that was analogous to pain-free controls. Brain responses demonstrated a significant within-group difference in FM patients, characterized by less brain activity bilaterally in the anterior insula following QR as compared to EX. There was also a significant Group by Condition interaction with FM patients showing less activity in the left dorsolateral prefrontal cortex following QR as compared to post-EX and CO following both conditions. These results suggest that exercise appeared to stimulate brain regions involved in descending pain inhibition in FM patients, decreasing their sensitivity to pain. Thus, exercise may benefit patients with FM via improving the functional capacity of the pain modulatory system.

## 1. Introduction

Fibromyalgia (FM) is a complex condition, characterized by chronic musculoskeletal pain along with a host of other interrelated symptoms. Although the mechanisms that maintain pain in FM are unknown, evidence points to a central nervous system dysregulation of pain processing. Specifically, research demonstrates heightened sensitivity to sensory stimuli, exaggerated brain responses to both painful and non-painful stimuli, and impairments in pain modulation [1].

The ability to modulate pain is critical for maintaining a functional balance between facilitation and inhibition of sensory stimuli and dysregulations in pain modulation can influence quality of life, disability and the development of chronic pain [2]. Exercise stimulates the pain modulatory system [3] and chronic aerobic exercise training is a consistently efficacious treatment for FM [4]. It is plausible then that exercise functions as a treatment by improving pain modulation. Previous research has examined the relationship between exercise and pain modulation in FM. However, current evidence is equivocal with some studies showing a decrease in pain sensitivity with exercise and others showing either no change or an increase [5,6]. Brain responses to pain following exercise remain largely unexplored in FM. A better understanding of these responses could help to explain the consequences of acute and chronic exercise for FM patients.

The purpose of the present study was to assess the influence of exercise on brain responses to pain in patients with FM and pain-free controls. Exercise was used as a method of stimulating pain modulatory (*i.e.*, central nervous) systems in FM and functional neuroimaging was employed to explore the neural responses underlying these processes.

## 2. Experimental Section

### 2.1. Participants

Female patients with a physician-confirmed diagnosis of FM who were between the ages of 18 and 60 years old, and age- and sex-matched pain-free controls (CO) were recruited from the community via advertisements. Exclusion criteria included: pregnancy, presence of ferrous metal in the body, left handedness, claustrophobia, medical conditions that would interfere with the ability to perform cycling exercise, diagnosis of major depressive disorder, and use of medications that would affect pain perception or the interpretation of brain data, including opioids, high-dose antidepressants and cardiovascular medications. Medication information and dosage were supplied by the patient and their physician and dosage levels (low, moderate, high) were determined through both physician consultation and use of the Physician’s Desk Reference [7]. A total of 12 female FM patients and 12 pain-free controls were enrolled in the study. Prior to testing, participants abstained from caffeine for 4 h, cigarettes for 2 h, alcohol for 24 h, structured exercise for 24 h and any pain medications for 24 h. Participants were compensated $100 for their time.

### 2.2. Procedures

The institutional review board at the University of Wisconsin-Madison approved the procedures of this study and informed consent was obtained from all participants. To characterize the sample, patients completed the Fibromyalgia Impact Questionnaire (FIQ) [8]. Participants underwent two fMRI scanning conditions, one following exercise (EX) and the other following quiet rest (QR). Order of conditions was randomized and counterbalanced. Perceptual and brain responses to a series of painful stimuli were collected during each scan. Self-reported pain symptoms were monitored pre and post scans with the Short Form—McGill Pain Questionnaire (MPQ) [9]. Participants’ scans were conducted at approximately the same time of day (± 1 h) and separated by a week. During the week between conditions, participants wore an ActiGraph accelerometer during waking hours to characterize regular physical activity and sedentary behaviors.

#### 2.2.1. Condition Descriptions

For the EX condition, participants completed 25 min of moderate intensity cycling on a Vision Fitness 2150, semi-recumbent stationary bicycle (Vision Fitness, Lake Mills, WI, USA). Following a 1-min warm-up, participants were instructed to achieve and maintain a pedaling rate of 60–70 revolutions per minute. They were also encouraged to increase or decrease the resistance level as needed to maintain a perception of effort that was “somewhat hard” or approximately “13” on Borg’s 6–20 ratings of perceived exertion (RPE) scale [10]. This intensity and duration of cycling was selected based on evidence from our group demonstrating that patients would be willing and able to complete the exercise [5]. Pedaling rate and RPE were monitored and recorded by research staff throughout cycling. For QR, participants rested on the same bike for the same amount of time as the exercise condition.

#### 2.2.2. fMRI Procedures

Three experimental runs, each including six heat pain stimuli (44 °C, 46 °C & 48 °C), were delivered to the left palm using a Medoc Pathway Thermal Sensory analyzer with a 573-mm^2^ CHEPS Peltier thermode. Experimental heat pain was chosen as our pain stimulus because many exercise and pain sensitivity studies have employed this modality and our primary interest was to explore brain responses to experimental pain stimuli. In this context, we are not testing clinical aspects of FM, but rather how the central nervous system regulates pain processes. Temperatures were each presented twice per run in a pseudorandom order; the order of temperatures was identical for each participant across the two conditions. No runs began with a 48 °C stimulus as previous research in our lab has shown that presenting the strongest temperature first has a significant effect on ratings for subsequent temperatures. Temperatures increased from 35 °C to the target temperature at a rate of 70 °C per second. Stimuli lasted 10 s followed by a 10-s rating period and a 20-s off period. High-resolution goggles and a button response device were used by participants to rate the intensity and unpleasantness of the heat pain using the Gracely Box Scales [11]. Each run lasted 4:28. For the scan post-exercise, the first run of pain stimuli was delivered on average 18:30 ± 1:45 min after participants stopped pedaling and for the scan post-quiet rest the time interval was 20:08 ± 2:27. These lags between exercise and scanning included the transition from the prep room to the scanner, instrumentation of participants for physiological monitoring, and preparatory scans (e.g., localizer). During scanning, heart rate (HR) and end-tidal CO_2_ (ETCO_2_) were recorded and included in statistical analyses to account for exercise-related changes in physiology that could influence the interpretation of the BOLD signal. Scanning procedures were identical on each testing day.

All functional and anatomical magnetic resonance images were collected on a 3-Tesla GE Discovery MR750 scanner (GE Health Systems, Waukesha, WI, USA). Following the localizer scan and high-order shimming, high-resolution functional T2* echo-planar blood oxygen level dependent (EPI-BOLD) images were obtained with an eight-channel transmit-receive head coil. Functional image acquisitions were obtained with a gradient echo sequence (TR 2000 ms, TE 25 ms, flip angle 60°) and consisted of 40 sagittal slices with thickness 4 mm and no gap, yielding coverage of the whole brain. The acquisition matrix was 64 × 64 and the FOV was 24 cm, delivering an in-plane voxel resolution of 3.75 mm × 3.75 mm × 4 mm. High-resolution T1-weighted anatomical acquisitions (TR 9 ms, TE 1.7 ms, FOV 24 cm, flip angle 10°) consisted of 128 axial slices with a matrix of 256 × 256, each 1.2 mm thick.

### 2.3. Data Processing and Analyses

Physical activity data used to characterize the sample were processed using standard procedures. Criteria for inclusion of accelerometer data were at least 10 h of valid wear time for a minimum of 3 weekdays and 1 weekend day. Non-wear time was defined as 60 min with zero acceleration in the vertical axis. In-house software was used to calculate minutes spent in sedentary, light, low-moderate, high-moderate, and vigorous levels of physical activity. Cut-off points for accelerometer counts per minute were based on previous research as follows: sedentary = 100 and below; light = 101–760; low-moderate = 761–1952; high-moderate = 1953–5724; vigorous = 5725 and above [12,13].

Functional brain imaging analyses were conducted using Analysis of Functional Neuroimages (AFNI) software [14,15] and all other statistical analyses were performed with SPSS 22.0 (SPSS Inc., Chicago, IL, USA). Anatomical images were registered to the Montreal Neurological Institutes (MNI) 152 template [16] using an affine transformation. For functional data, the initial five time points were discarded from functional analyses due to saturation effects. Data were motion corrected (3dvolreg), orthogonalized to periodic components of the physiological signal (IRF-RETROICOR from PESTICA [17,18]), de-spiked (3dDespike), slice-time corrected (3dTshift), aligned to the MNI-152 template with a nonlinear warp (ANTS WarpTimeSeriesImageMultiTransform), iteratively blurred to a smoothness of 11.3 mm full-width, half-maximum (3dBlurToFWHM) and converted to percent signal change. AFNI’s 3dREMLfit program was used to perform linear regression on each participant’s data including separate regressors for the pre-stimulus countdown, heat stimulus, and rating period.

#### 2.3.1. Corrections for Physiological Noise

To control for longer-term physiological effects, in each subject’s final model we included HR and ETCO_2_ regressors convolved with five tent functions, evenly spaced from 0 s to 20 s (*i.e.*, at 0, 5, 10, 15 and 20 s). This was done in order to model the unknown hemodynamic responses to these physiological processes, which have higher latencies and longer durations than responses to the types of tasks that are often performed in the neuroimaging environment (e.g., responses to cognitive tasks) [19,20]. We elected to estimate these hemodynamic responses for each subject anew rather than use the estimates of Birn [19] and Chang [20] because of the variation of the hemodynamic response to different types of changes in respiration (e.g., breath holding, cued depth and rate changes, free breathing) observed previously [19], which suggested that the hemodynamic responses estimated in those references might not generalize to subjects immediately post-exercise. Thus, while we chose an approach to correct for physiological noise similar to these methods, we are not employing their exact correction.

#### 2.3.2. Group Level Analyses

Group-level brain imaging analyses were limited to regions of interest (ROIs) based on research in the areas of central nervous system processing of pain stimuli [21]. Regions included pre- and postcentral gyri, superior parietal lobule, cingulate cortices, brainstem, frontal medial cortex, frontal and parietal opercula, frontal pole, insula, thalamus, and middle frontal and orbital frontal gyri.

Pain intensity and unpleasantness ratings (0–20) were averaged across presentations of stimuli at each temperature within each run and were compared using Group (FM & CO) by Condition (EX & QR) by Run (1, 2 & 3) by Temperature (44 °C, 46 °C & 48 °C) repeated measures ANOVAs with the primary contrasts of interest being the main effects for Group and Condition and the interaction of these variables. We observed order effects for the CO group, such that pain ratings to experimental pain were generally higher on Day 1 compared to Day 2, regardless of condition (EX or QR). Thus, we included the Condition order as a covariate in our repeated measures ANOVAs. Simple effects were used to further explore significant results across Run and Temperature. Effect sizes (Cohen’s *d*) were used to characterize the magnitude of the differences in pain ratings between conditions. Paired-samples *t*-tests and effect sizes were also used to compare pain symptoms (MPQ visual analog scale) pre and post-scanning for FM patients.

Brain responses to pain were analyzed using AFNI’s Linear Mixed Effects (3dLME) program with condition and group as the independent variables and brain responses to pain as the dependent variable. To control for multiple comparisons, we thresholded the statistical map at a voxelwise *p*-value of 0.01 and applied a cluster-size threshold of 43 voxels (2752 mm^3^), corresponding to a cluster-wise alpha of 0.05 as determined by AFNI’s 3dClustSim. Six Pearson correlation coefficients were used to examine the relationships between differences in brain responses to pain (for significant regions of interest only) and differences in self-reported experimental pain ratings (intensity and unpleasantness) between EX and QR. To control for multiple comparisons, three families of correlations were created, one for each region, and a Bonferroni correction was applied reducing the alpha level for significance for these analyses to 0.025.

Questionnaire and physical activity data were used to characterize the sample; group comparisons of these data were performed using independent samples *t*-tests. Physiological data collected in the scanner (HR, ETCO_2_) were averaged across each run and compared within and between groups using repeated measures ANOVAs. Alpha was set to 0.05 for these comparisons.

## 3. Results and Discussion

One FM patient was unable to return for exercise due to scheduling complications so data from 11 patients and 12 controls are presented (see Table 1 for participant characteristics).

### 3.1. Influence of Exercise on Pain Ratings and Symptoms

Detailed results for pain intensity (PI) and pain unpleasantness (PU) ratings during each run are presented in Table 2 and illustrated (Run 1 only) in Figure 1. For both PI and PU ratings, there were significant Main Effects for Temperature (PI: *p* = 0.002; PU: *p* = 0.010) and Condition (PI: *p* = 0.005; PU: *p* = 0.012) and significant Condition by Run interactions (PI: *p* = 0.032; PU: *p* = 0.030). The Condition by Run interaction indicated that pain ratings differed between conditions as a function of run. Neither the Main Effect of Group nor the Group by Condition interaction were significant (*p* > 0.05).

Further examination of the significant interactions using effect size estimates indicated that the results were largely driven by higher pain ratings during the QR condition in FM patients (see Figure 1). Specifically for FM patients, PI and PU ratings were moderately higher during the first run post-QR as compared to the first run following EX (ΔPI: 2.0, 95% CI (0.9, 5.0); ΔPU: 1.8, 95% CI (0.6, 4.2)), demonstrating a small and transient hypoalgesic effect for EX (*d* = 0.39–0.41). Ratings were comparable for FM patients between EX and QR for runs 2 and 3 (*d* = 0.07–0.14). Effect size estimates for CO showed that pain rating differences between EX and QR were consistently small (*d* ≤ 0.20) across runs. Moreover, effect size estimates for group comparisons demonstrated that pain ratings post-EX in FM were similar to those seen in controls during the first pain run (*d* = 0.10–0.26; *p* > 0.05) but showed small to moderate elevations compared with controls during the second and third runs as the effects of exercise diminished (*d* = 0.30–0.65). Similarly, following QR, FM patients’ ratings were meaningfully elevated compared to CO (*d* = 0.51–0.93).

Following the scan post-EX, FM patients experienced a decrease in pain symptoms as reported from the MPQ visual analog scale (pre = 57.8 ± 29.1; post = 53.1 ± 27.6; *d* = 0.39) whereas after the QR scan there was an increase in pain symptoms (pre = 50.5 ± 18.2; post = 57.3 ± 20.5; *d* = 0.17). Following EX, HR was significantly elevated (*p* < 0.001) and ETCO_2_ was significantly lower across runs (*p* = 0.013) compared to QR. Groups did not differ significantly in HR (*p* > 0.05) and there were no significant interactions (*p* > 0.05). However, ETCO_2_ was significantly higher in FM compared to CO across runs (*p* < 0.001).

### 3.2. Influence of Exercise on Brain Processing of Pain

For neuroimaging data five participants were excluded (2FM, 3CO) due to excessive head movement (>2 mm, *n* = 3) and missing HR data (*n* = 2) reducing the sample size for these analyses to nine individuals per group. There was a significant within-group difference in FM patients, characterized by greater brain activity bilaterally in the anterior insula following EX as compared to QR (see Figure 2; *p* < 0.01). In CO, there was also a significant within-group interaction, characterized by greater activity in the right parietal operculum and the right pre/postcentral gyrus following exercise as compared to quiet rest (*p* < 0.05). Further, there was a significant Group by Condition interaction (*p* < 0.05) characterized by relatively greater activity in the left dorsolateral prefrontal cortex (DLPFC) for FM patients following EX compared to both the QR condition and CO (both conditions; see Figure 3). There were no additional significant group differences (*p* > 0.05) in brain responses to pain between FM or CO for either EX or QR conditions. Volumes, *t*-statistics and coordinates for significant regions are shown in Table 3.

### 3.3. Relationship between Brain Responses and Pain Ratings

For the DLPFC, there was a significant relationship between changes in brain responses between EX and QR and changes in pain intensity ratings across groups between the two conditions (*r* = −0.32, *p* = 0.02). The correlation between brain responses in this region and changes in PU was similar in magnitude and direction, but failed to reach significance (*r* = −0.25, *p* = 0.06). For the left and right anterior insula, correlation coefficients were similar in magnitude to those for the DLPFC, but did not reach significance for either PI (R Ins: *r* = −0.29, *p* = 0.14; L Ins: *r* = −0.28, *p* = 0.16) or PU ratings (R Ins: *r* = −0.20, *p* = 0.33; L Ins: *r* = −0.30, *p* = 0.15).

## 4. Discussion

This study examined brain responses to pain post-exercise in FM patients and pain-free controls using functional neuroimaging. Our results demonstrated that a short bout of moderate intensity cycling resulted in temporary improvements in centrally mediated pain modulation in FM patients. Notably, both brain responses and pain ratings for FM patients were similar to those of controls immediately post-exercise. However, pain ratings for FM were lower following exercise compared to quiet rest and this was accompanied by an increase in activity among brain regions involved in pain modulation [1,22]. We also observed a significant relationship between changes in DLPFC brain activity and changes in pain perception. Overall, these results suggest that a moderate intensity bout of acute exercise can influence descending regulatory systems in FM. Considering that exercise has been shown to be an efficacious treatment for FM, these results suggest that symptom improvements in FM following exercise training may be, in part, the result of increases in the functional capacity of the pain modulatory system.

Differences were seen in the anterior insula comparing exercise to quiet rest for FM patients and there were differences both within and between groups in the DLPFC, brain regions that are strongly implicated in pain modulation [21,22]. The anterior insula has been classified as a primary component of the pain salience network [23] and several studies have highlighted the insula as a critical brain region discriminating FM from healthy controls [22]. Activity in the DLPFC is also strongly implicated in pain modulation [24] and previous data from our group demonstrated that activity in this region was negatively associated with sedentary behavior and pain modulation in FM patients [25]. This suggests that modifiable behaviors such as prolonged sitting and/or physical activity may impact pain modulation. Our present results are in-line with those demonstrating that following quiet rest, which likely represents typical neuroimaging conditions, patients had decreases in activity in this region in comparison to baseline. Further, following exercise, patients showed small increases in brain activity from baseline; a response that was consistent with brain responses in healthy controls at rest.

Previous research regarding the effects of acute bouts of exercise on pain sensitivity in FM patients is largely equivocal. Some studies have demonstrated a hypoalgesic effect [5,26] while other studies show either no changes in pain perception or an exacerbation of pain [27,28,29]. With respect to exercise training, the results are more consistent and a large body of evidence now demonstrates that regular exercise is beneficial for FM [4]. However, the mechanisms that drive these benefits are largely unknown. Our results suggest that moderate intensity exercise can stimulate central pain regulatory mechanisms. Notably, we found that exercise results in beneficial alterations of both pain perception a brain responses to pain for approximately 20–30 min post-exercise. While temporary, these improvements may become more permanent with repeated exposure (*i.e.*, a positive adaptation to chronic exercise) and may provide mechanistic insight into the efficacy for exercise training in the treatment of FM. In support of this contention is recent evidence that six weeks of aerobic exercise training increased ischemic pain tolerance in healthy individuals [30] and that physical activity is related to pain modulation in both healthy individuals and FM [31,32]. How these results translate to exercise training in patients with chronic pain is currently unknown. Clinical trials determining whether exercise training can strengthen central pain modulation and whether these changes are associated with symptom improvements are needed to more fully test this hypothesis.

There was also a difference in brain responses for controls between exercise and quiet rest conditions in the right parietal operculum and in an area spanning the right pre- and postcentral gyri, areas contralateral to the pain stimulus. These regions are consistently active in response to nociceptive stimuli and are known to have somatotopic organization suggesting that they encode spatial aspects of pain processing [21,33]. Controls had greater activity in these regions post-exercise compared to quiet rest demonstrating that exercise influenced these areas. Previous research has also demonstrated their involvement in pain post-exercise as a function of delayed onset muscle soreness [34]. However, pain ratings did not differ between conditions in controls making a conclusive interpretation of these results difficult. Future studies in healthy individuals where exercise intensity is high enough to induce a hypoalgesic response will be critical for the further exploration of the influence of exercise on a typically-functioning central nervous system.

This study had a number of limitations. We included a small sample of patients and controls and our sample included only women, limiting the generalizability of our findings. Further, we excluded individuals with diagnosed mental health conditions and those who were taking medications that could impact pain or the interpretation of brain responses. As such, our results may not apply to FM patients with comorbid conditions. However, our group of patients had FIQ scores and levels of physical activity and sedentary behaviors that are comparable to those previously reported in FM patients [8]. Further, we intentionally chose a moderate intensity exercise stimulus in order to ensure that our patients would be able to complete the exercise protocol and chose to keep the relative intensity consistent between groups to improve comparability. However, it appears that the exercise bout as prescribed was not a sufficient stimulus to induce a decrease in pain sensitivity in our pain-free controls. This is consistent with previous literature regarding the effects of acute exercise on pain in healthy controls, which typically finds that higher intensity exercise is necessary to elicit a hypoalgesic response [35]. Thus, we were unable to compare brain responses during exercise-induced hypoalgesia between groups.

Our study also had a number of notable strengths. Most importantly, we used functional neuroimaging to examine brain responses to pain relatively soon post-exercise allowing us to explore the acute effects of exercise on central nervous system responses. As part of this process, we measured and statistically controlled for the physiological effects of exercise that could impact the BOLD response. Though the evidence is equivocal, it has been suggested that cardiovascular mechanisms may be involved in the hypoalgesic response to exercise [36] and, thus statistically controlling for these effects may have influenced the interpretation of our results. However, without controlling for physiological differences between EX and QR the data would have been difficult to interpret. Lastly, we used an objective measure to assess physical activity to confirm that our sample had typical levels of physical activity for this population [37,38].

A growing number of studies have begun to employ neuroimaging methods to better understand the impact of exercise on the brain both longitudinally and acutely. For example, Smith and colleagues [39] conducted fMRI scans before and after an exercise training program in older adults with mild cognitive impairment and found that exercise improved neural efficiency during cognitive tasks post-intervention. Structural MRI has also been used to show the neuroprotective effects of regular exercise in older adults with respect to preservation of brain volume [40]. In contrast to using neuroimaging to track changes in the brain over time, neuroimaging during and immediately following exercise presents some unique challenges due to artifacts associated with movement and the physiological underpinnings of many neuroimaging methods (e.g., BOLD response). EEG has been used most extensively to explore the effects of exercise on cortical activity [41]. PET and fMRI have also been used, though to a much lesser extent. For example, Boecker and colleagues [42] used PET to demonstrate the effects of a long-distance run on opioid release in the brain and Janse Van Rensberg and colleagues [43,44] used fMRI to examine brain responses to nicotine craving following 10 min of moderate intensity cycling. Our study adds to this important body of literature by using fMRI to show that an acute bout of moderate intensity exercise improved brain mechanisms underlying pain modulation in patients with chronic pain and further highlights the potential benefits of utilizing neuroimaging technology to better understand the more immediate effects of exercise on the human brain.

## 5. Conclusions

In summary, we present a novel view of the impact of exercise on the central nervous system’s involvement in pain modulation as well as mechanisms underlying the pain-relieving effects of exercise in FM. Our results demonstrated that a relatively short bout of cycling exercise resulted in improvements in pain modulation in FM. This research may lead to future studies aimed towards determining whether the therapeutic effects of exercise training result from changes in central pain regulatory mechanisms. Research examining these effects in FM using functional brain imaging methods in the context of an exercise intervention trial will be necessary to more fully test this hypothesis.

## Figures and Tables

**Figure 1 brainsci-06-00008-f001:**
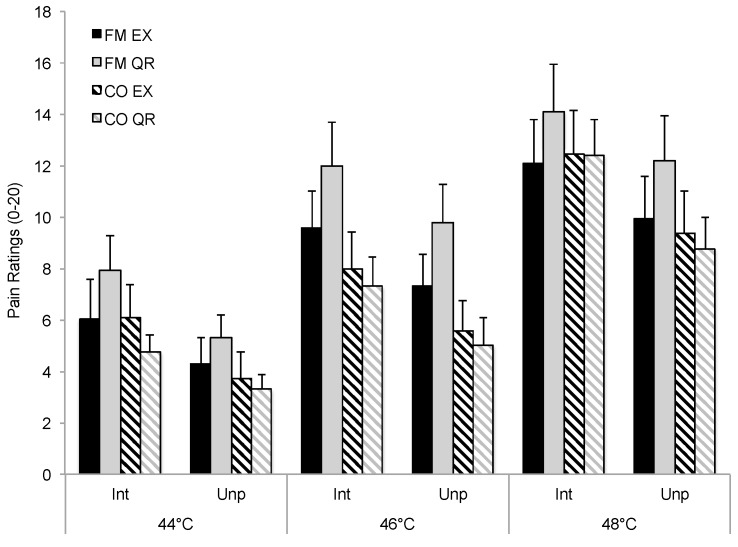
Pain intensity (Int) and unpleasantness (Unp) ratings during the first run of each condition. FM = fibromyalgia; CO = control; EX = post-exercise; QR = post quiet rest.

**Figure 2 brainsci-06-00008-f002:**
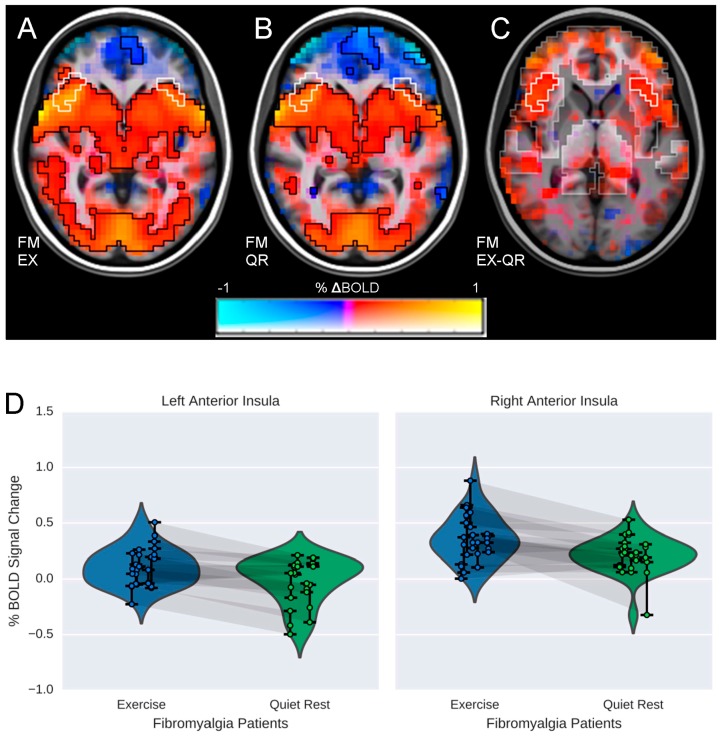
Maps of BOLD responses to pain in nine FM patients. (**A**) BOLD response post-exercise (EX); (**B**) BOLD response post-quiet rest (QR); (**C**) within group differences in BOLD responses in the bilateral anterior insula (EX-QR). Color represents the *β* coefficient (% signal change ranging from −1% to 1%) and opacity represents *t*-statistic, with full opacity at a voxelwise *t* corresponding to *p* < 0.01. The significant cluster at *α* < 0.05 (corresponding to a cluster size threshold of 17 voxels) is outlined in white. Analyses were performed at 4 mm × 4 mm × 4 mm resolution using a mask of regions determined *a priori* from the hypotheses, which is highlighted in the background of image C; (**D**) Violin plots illustrating condition differences in %BOLD signal change in the left and right anterior insulae. Each point represents the average %∆BOLD in the cluster during one run, so that each subject is represented by three connected points. Shadows connecting points between violin plots indicate data from the same individual during each condition (EX & QR). Note that the region from which these points are drawn was chosen because there is a significant difference between EX and QR; the plot is intended to help clarify the within-subjects’ results.

**Figure 3 brainsci-06-00008-f003:**
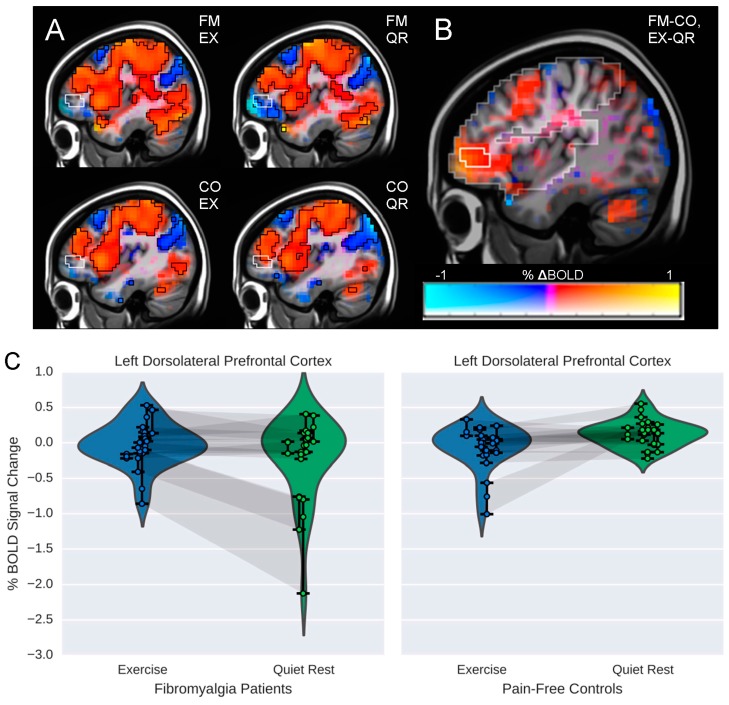
Maps of BOLD responses to pain in 9 fibromyalgia (FM) patients and nine pain-free controls (CO). (**A**) BOLD response by group and condition (EX = exercise; QR = quiet rest); (**B**) significant group by condition interaction in BOLD responses in the left dorsolateral prefrontal cortex (DLPFC). Color represents the *β* coefficient (% signal change ranging from −1% to 1%) and opacity represents *t*-statistic, with full opacity at a voxelwise *t* corresponding to *p* < 0.01. The significant cluster at *α*<0.05 (corresponding to a cluster size threshold of 17 voxels) is outlined in white. Analyses were performed at 4 mm × 4 mm × 4 mm resolution using a mask of regions determined *a priori* from the hypotheses, which is highlighted in the background of image B; (**C**) Violin plot illustrating the simple main effects underlying the interaction in the DLPFC. Each point represents the average %∆BOLD in the indicated cluster during one run, such that each subject is represented by three connected points. Shadows connecting points between violin plots indicate data from the same individual during each condition (EX & QR). Note that the region from which these points are drawn was chosen because there is a significant interaction between group and condition; the plot is intended to further clarify the interaction.

**Table 1 brainsci-06-00008-t001:** Demographics, pain symptoms, and physical activity data.

		FM (*n* = 11) Mean (SD)	CO (*n* = 12) Mean (SD)
Age (years)		38.58 (11.17)	43.67 (7.02)
Height (cm)		165.18 (6.77)	165.92 (5.77)
Weight (kg)		65.55 (11.17)	68.41 (10.20)
FIQ		52.95 (13.06)	NA
MPQ VAS	Pre-Exercise	57.82 (29.10) *	4.83 (10.18)
	Pre-Quiet Rest	50.50 (18.24) *	1.33 (2.77)
Physical Activity Data (minutes/day)	Sedentary	624.40 (78.69)	625.88 (81.35)
	Light	164.77 (48.17)	158.43 (37.42)
	Moderate	82.15 (31.23)	108.04 (39.8)
	Vigorous	0.28 (0.55)	3.58 (9.27)

* FM significantly greater than CO, *p* < 0.05; FIQ = Fibromyalgia Impact Questionnaire; FM = Fibromyalgia; CO = control.

**Table 2 brainsci-06-00008-t002:** Pain intensity (PI) and unpleasantness (PU) ratings (Mean (SD)) for pain delivered post-exercise (EX) and post quiet rest (QR).

		FM Patients (*n* = 11)		CO (*n* = 12)	
		PI	PU	PI	PU
EX	Run 1	9.31 (5.07)	7.30 (4.36)	8.83 (4.42)	6.24 (3.74)
	Run 2	10.03 (4.49)	7.97 (4.14)	8.51 (4.06)	5.42 (3.73)
	Run 3	9.88 (4.86)	7.80 (4.39)	8.47 (4.60)	5.76 (3.70)
QR	Run 1	11.33 (5.21)	9.09 (4.30)	8.15 (3.44)	5.68 (3.01)
	Run 2	9.60 (3.96)	7.47 (3.21)	7.31 (3.50)	4.96 (2.39)
	Run 3	9.50 (3.60)	7.53 (3.40)	7.79 (3.16)	5.35 (2.40)

**Table 3 brainsci-06-00008-t003:** Results from the Linear Mixed Effects Model examining the differences in brain responses between groups (Fibromyalgia (FM), Control (CO)) and sessions (Exercise (EX) and Quiet Rest (QR)).

	Direction	Peak X,Y,Z	Volume (mm^3^)	Peak *t*-Statistic	α
**FM EX-QR**					
Left Anterior Insula	+	32, −20, −18	4608	2.83	<0.01
Right Anterior Insula	+	−52, −20, −6	4032	2.88	<0.01
**CO EX-QR**					
Right parietal operculum	+	−64, 28, 14	4416	5.32	<0.01
Right pre/postcentral gyrus	+	−36, 24, 50	2816	3.57	<0.05
**FM-CO × EX-QR**					
Left DLPFC	+	28, −56, −14	3136	2.91	<0.03

Included in the table are clusters showing differences between groups and sessions. For each analysis, multiple comparisons were corrected for using a cluster-size threshold of 43 voxels or 2752 mm^3^.

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
