# Peer review of "Exercise Strengthens Central Nervous System Modulation of Pain in Fibromyalgia"

_brainsci, 2016, doi:10.3390/brainsci6010008_

Round 1
Reviewer 1 Report
This manuscript describes a cohort randomized crossover design comparing cutaneous heat pain ratings and brain activity during heat stimuli after conditions of quiet rest and cycling in patients with a physician-confirmed diagnosis of Fibromyalgia (FM) and age and sex matched controls. The methodology is sound and the manuscript is well-written. The discussion interprets the results as supporting evidence of exercise-induced hypoalgesia (EIH) in the patients with FM. However, the authors need to compare their findings to other research evidence showing impaired EIH and/or descending inhibition in patients with FM - especially considering that their results did not support EIH in the controls. The authors merely mention that the exercise stimulus may have been insufficient for the controls to show EIH without critically discussing what such a conclusion would mean (eg, FM patients have better EIH mechanisms than the controls). Also, the discussion section should be revised to acknowledge that many patients with FM would have been excluded from participation in this study due to major depression and medication consumption so there may be concerns about the generalizability of the sample, BUT the authors appropriately compared the Fibromyalgia Impact Questionnaire scores of their sample to normative data and found no significant difference. In addition, the authors might want to consider and mention that their method of controlling “physiological noise” might have influenced the results because cardiovascular reactions to exercise were, at one time, proposed to be a mechanism of EIH.
Specific Comments:
· P. 1, L. 40 – I don’t see that the reference actually includes studies of activities of daily living so the authors should revise the sentence or add an appropriate reference.
· P. 2, L. 18 – A reference for the determination that a dosage of antidepressants was “high” should be provided.
· P. 3, L. 2 – Each temperature was applied twice so I assume the averages were used, but the authors should clarify.
· P. 3, L. 15 – The time post-exercise before scanning was provided, but not the time post-quiet rest.
· P. 6, L. 5 – “Elevations” should be revised to “higher” so that readers do not erroneously believe that a pre-scan application of heat stimuli was administered.
· P. 6, L. 12 - I am uncertain why the authors only report the effect sizes of group differences for the first run.s
Author Response
We would like to thank the reviewer for their positive and critical review of our work. We have addressed each concern below and made changes in the manuscript accordingly. We feel that the manuscript has been significantly improved as a result of the suggested revisions. Our responses to the reviewer’s specific concerns follow each point and are indented and in red font.
We concur that a brief description of the literature regarding EIH in FM is warranted and have added information to the discussion as follows: “Previous research regarding the effects of acute bouts of exercise on pain sensitivity in FM patients is largely equivocal. Some studies have demonstrated a hypoalgesic effect [5,24] while other studies show either no changes in pain perception or an exacerbation of pain [25–27].”
With respect to the reviewer’s comment regarding the lack of EIH in controls, we have included some additional information to demonstrate the consistency of our findings with previous research. The passage now reads: “However, it appears that the exercise bout as prescribed was not a sufficient stimulus to induce a decrease in pain sensitivity in our pain-free controls. This is consistent with previous literature regarding the effects of acute exercise on pain in healthy controls, which typically finds that higher intensity exercise is necessary to elicit a hypoalgesic response [33].” Additionally, we want to again mention that our study was not primarily designed to induce EIH in controls; it was designed to examine brain responses to pain following exercise and thus we wanted to ensure that FM patients could complete the exercise bout. Exploring brain responses associated with EIH in healthy individuals would be an interesting future direction.
With respect to the reviewer’s comment on generalizability, we agree that our exclusionary criteria were stringent and have added the following sentence to our limitations section. “Further, we excluded individuals with diagnosed mental health conditions and those who were taking medications that could impact pain or the interpretation of brain responses. As such our results may not apply to FM patients with comorbid conditions.”
Lastly, with respect to the comment about controlling for physiological noise, the following statements have been added. “Though the evidence is equivocal, it has been suggested that cardiovascular mechanisms may be involved in the hypoalgesic response to exercise [34] and thus statistically controlling for these effects may have influenced the interpretation of our results. However, without controlling physiological differences between EX and QR the neuroimaging data would have been very difficult to interpret.”
I don’t see that the reference actually indicates studies of activities of daily living so the authors should revise the sentence or add an appropriate reference.
This sentence has been revised to better reflect the reference used.
A reference for the determination that a dosage of antidepressants was “high” should be provided.
We agree. We consulted with each physician with respect to type and dosage of medication, including whether this was a low, moderate or high dose. We also double-checked each medication and dose with the Physician’s Desk Reference (60th-65th editions). We have added the following information to the methods section “Medication information and dosage were supplied by the patient and their physician and dosage levels (low, moderate, high) were determined through both physician consultation and use of the Physician’s Desk Reference (65th edition). “
Each temperature was applied twice so I assume averages were used, but the authors should clarify.
This addition has been made.
The time post-exercise before scanning was provided, but not the time post-quiet rest.
This addition has been made.
Elevations should be revised to higher so that readers do not erroneously believe that a pre-scan application of heat stimuli was administered.
The suggested change has been made.
I am uncertain why the authors only report the effect sizes of group differences in the first run.
Thank you for catching this omission. This was an oversight on our part and these effect sizes are now included on page 11.
Reviewer 2 Report
This study has compared acute effects of exercise and inactivity in fibromyalgia (FM) and pain-free controls on changes in pain and cerebral activity in response to heat.
Following exercise in FM patients, activity was transiently increased in anterior insula and dorsolateral prefrontal cortex (DLPFC) while activity was transienly decreased following rest. In paralell pain sensitivity to heat was decreased by exercise.
Changes in pain sensitivity after exercise versus rest were significantly correlated with changes in activity in DLPFC (exercise vs. rest). However, in the latter case the authors need to address the issue of multiple correlations.
Nine individuals in each group were included in neuroimaging analyses, this should be indicated in the abstract.
In table 3 the subheading "Peak X, Y, X" needs correction.
Consider discussing the results in relation to previous studies on exercise and neuroimaging
The results are clearly reported and adequately discussed. The findings are novel and may be compared to previous studies of exercise and neuroimaging in fibromyalgia.
I will be happy to read the revised version of the manuscript.
Author Response
We would like to thank the reviewer for their positive and critical review of our work. We have addressed each concern below and made changes in the manuscript accordingly. We feel that the manuscript has been significantly improved as a result of the suggested revisions. Our responses to the reviewer’s specific concerns follow each point and are indented and in red font.
Changes in pain sensitivity after exercise vs. quiet rest were significantly correlated with changes in activity in the DLPFC (exercise vs. rest). However, in the latter case the authors need to address the issue of multiple correlations.
These analyses were intended primarily for descriptive purposes. However, we recognize the need to control for multiple comparisons in order to reduce the risk of making Type I errors. Therefore, we created 3 families including 2 correlations each (one for each of the significant regions) and performed a Bonferroni correction, making the critical alpha level for significance 0.025.
This has been clarified in the statistical analysis section.
Nine individuals in each group were indicated in the neuroimaging analysis, this should be indicated in the abstract.
This change has been made.
In table 3, the subheading “Peak X, Y, X” needs correction.
This correction has been made.
Considering discussion the results in relation to previous studies on exercise and neuroimaging.
We agree that this would add valuable information to the discussion section and have added the paragraph shown below discussing previous work using neuroimaging to understand the effects of exercise on the brain.
A growing number of studies have begun to employ neuroimaging methods to better understand the impact of exercise on the brain both longitudinally and acutely. For example, Smith and colleagues [37] conducted fMRI scans pre and post an exercise training program in older adults with mild cognitive impairment and found that exercise improved neural efficiency during cognitive tasks post-intervention. Structural MRI has also been used to show the neuroprotective effects of regular exercise in older adults with respect to preservation of brain volume[38]. In contrast to using neuroimaging to track changes in the brain over time, neuroimaging during and immediately following exercise presents some unique challenges due to artifacts associated with movement and the physiological underpinnings of many neuroimaging methods (e.g. BOLD response). EEG has been used most extensively to explore the effects of exercise on cortical activity [39]. PET and fMRI have also been used, though to a much lesser extent. For example, Boecker and colleagues used PET to demonstrate the effects of a long-distance run on opioid release in the brain and Janse Van Rensberg and colleagues used fMRI to examine brain responses to nicotine craving following 10 minutes of moderate intensity cycling. Our study adds to this important body of literature by using fMRI to show that an acute bout of moderate intensity exercise improved brain mechanisms underlying pain modulation in patients with chronic pain and further highlights the potential benefits of utilizing neuroimaging technology to better understand the more immediate effects of exercise on the human brain.
Round 2
Reviewer 1 Report
The authors have adequately addressed my concerns. I just see a grammatical error on page 13 line 11 where the manuscript reads "there were also a difference...."